# Study on Metronidazole Acid-Base Behavior and Speciation with Ca^2+^ for Potential Applications in Natural Waters

**DOI:** 10.3390/molecules27175394

**Published:** 2022-08-24

**Authors:** Federica Carnamucio, Claudia Foti, Massimiliano Cordaro, Ottavia Giuffrè

**Affiliations:** 1Dipartimento di Scienze Chimiche, Biologiche, Farmaceutiche ed Ambientali, Università di Messina, Viale F. Stagno d’Alcontres 31, 98166 Messina, Italy; 2CNR-ITAE, Istituto di Tecnologie Avanzate per l’Energia, 98126 Messina, Italy

**Keywords:** metronidazole, protonation constants, Ca^2+^ complexes, speciation, sequestration, potentiometry, ^1^H NMR spectroscopy, UV-Vis spectroscopy, thermodynamic parameters

## Abstract

Metronidazole (*MNZ*) is an antibiotic widely used for the treatment of various infectious diseases and as an effective pesticide agent for the cultivation of chickens and fish. Its high resistance to purification processes and biological activity has led to the classification of *MNZ* as an emerging contaminant. A speciation study, aimed to define the acid-base properties of *MNZ* and its interaction with Ca^2+^, commonly present in natural waters, is reported. The protonation constants of *MNZ,* as well as the formation constant value of Ca^2+^-*MNZ* species, were obtained by potentiometric titrations in an aqueous solution, using NaCl as background salt at different ionic strengths (0.15, 0.5, 1 mol L^−1^) and temperature (15, 25 and 37 °C) conditions. The acid-base behavior and the complexation with Ca^2+^ were also investigated by ^1^H NMR and UV-Vis titrations, with results in very good agreement with the potentiometric ones. The dependence of the formation constants on the ionic strength and temperature was also determined. The sequestering ability of *MNZ* towards Ca^2+^ was defined by the empirical parameter pL_0.5_ at different pH and temperature values. The speciation of *MNZ* simulating sea water conditions was calculated.

## 1. Introduction

Metronidazole (2-Methyl-5-nitroimidazole-1-ethanol, *MNZ*) is a synthetic nitroimidazole widely employed in human and veterinary medicine as an antibiotic drug for its broad-spectrum antibacterial and antiprotozoal activity [1,2,3,4]. It represents one of the most used drugs around the world [5], also exploited as an additive to remove parasites in poultry and fish feed. Its widespread use has led to its classification as an emerging organic contaminant (CEC) [6]. *MNZ*, characterized by a ring-like structure, shown in Figure 1, is known as carcinogenic to animals, and as a potential human carcinogen and mutant [7,8,9,10].

In the literature, the use of different techniques was reported for separation or quantitative *MNZ* detection in environmental and biological samples, human plasma, drugs, and food products. They include high-performance liquid chromatography [11], liquid chromatography-tandem mass spectrometric method (LC-MS/MS) [12], chemiluminescence [13] fluorescence [14,15], near-infrared spectrophotometry (NIRS) [16], spectrophotometry [17,18,19], voltammetry with chemically modified electrodes [20,21,22,23,24,25,26,27,28,29].

*MNZ* belongs to pharmaceutical active compounds (PhAC), its accumulation in natural waters due to improper disposal of pharmaceutical waste represents a threat to the environment and to humans due to its high solubility in water, limited biodegradability, photo and hydrolytic stability, and toxicity [30,31]. Therefore, it is particularly relevant to explore effective methods for the removal and degradation of *MNZ* from several environmental sources, including natural waters. Among the various techniques, adsorption, oxidation processes, ultraviolet light degradation, biological methods, and thermocatalytic activation, were employed to remove antibiotics from polluted waters [11,29,32,33,34,35,36]. Adsorption methods are promising as they are widely accepted as an economic, benign, and easily applicable technique [37]. As is well known, the sorption processes were strictly dependent on pH, therefore preliminary studies on acid-base properties of *MNZ* under different conditions are crucial for this kind of application. 

Moreover, in natural waters, wherein *MNZ* is present, other inorganic components, such as metal cations and anions are also widely available. The main natural water constituents influence removal efficiency, depending on the properties of both the pollutants and components. For example, bivalent cations, such as Ca^2+^, and Mg^2+^, have a more negative effect than monovalent cations (Na^+^, K^+^) [38]. Calcium is the fifth most abundant element in the environment as well as in the human body. It is an essential element for life, playing relevant biological roles in cellular, molecular, and systemic processes in vertebrates. It is one of the principal ions in sea water, together with Na^+^, K^+^, Mg^2+^, Cl^−^, and SO_4_^2−^. Calcium ion is a large divalent cation widely available in most natural waters as its common salts are not very insoluble. In the oceans, its content is an average of 390 mg kg^−1^, with higher values of 440 mg kg^−1^ close to riff regions [39]. Rivers present quite different calcium contents, depending on their geological origin. As an example, in European rivers calcium content varies from 3.6 mg L^−1^ in Norwegian rivers to 84 mg L^−1^ in the lower polluted Rhine River [39]. By considering all these reasons, i.e., the presence of *MNZ* in natural waters and its potential interactions with the most abundant cations in sea waters, the purpose of this study is to elucidate the acid-base behavior and the speciation of *MNZ* with Ca^2+^. The results of the investigation can find applications in improving *MNZ* removal strategies.

## 2. Results and Discussion

The knowledge of the protonation constants of a ligand is crucial to defining the ionization state which influences several characteristics, such as aqueous solubility, and complexation ability. The equilibrium reactions associated with the protonation constant values are the following, where the charges are omitted for simplicity and *MNZ* was indicated as L:
iH + L = H_i_L      β^H^^iL^(1)
H + H_i−1_L = H_i_L     *K*^H^^iL^(2)

The first step involves the protonation of the alcoholic group, and the second step refers to the imidazole residue, as indicated in the paper of Rivera-Utrilla et al. [40]. The experimental protonation constant values obtained by potentiometry, at different ionic strengths (0.15, 0.5, 1 mol L^−1^) and temperature (15, 25, and 37 °C) conditions, are listed in Table 1.

As can be observed in the distribution diagram in Figure 2a, at *t* = 25 °C and *I* = 0.15 mol L^−1^, the mono-protonated species is formed in all pH ranges, reaching its maximum ligand fraction, corresponding to over 0.9, in the range 4 ≤ pH ≤ 10. Differently, the fractions of the di-protonated and deprotonated species became significant at very acid pH (pH = 2) and alkaline pH values (pH = 11), respectively.

To investigate the interaction between *MNZ* and Ca^2+^, various potentiometric titrations were also performed on solutions containing different metal/ligand ratios. The reactions related to formation constants are indicated below, where the charges are omitted for simplicity and *MNZ* was indicated as L:Ca + L + rH = MLH_r_    β^MLH^^r^(3)
Ca + LH_r_ = MLH_r_     *K*^MLH^^r^(4)

Hydrolysis constant values of Ca^2+^ species at various temperatures and ionic strength in NaCl, taken into account in the calculations relating to the M-L system, are reported in Appendix A. The most accurate speciation model is chosen by evaluating several factors, which are the statistical parameters standard and mean deviation, the formation percentages of the considered species, and the simplicity of the model [41]. Based on potentiometric results, several computational tests were carried out considering different speciation models. The only model that produced results at all investigated ionic strengths and temperatures is very simple, including the formation of CaLH species, whose values of formation constants are listed in Table 1.

More in detail, at 25 °C and *I* = 0.15 mol L^−1^, as shown in the speciation diagram in Figure 2b, the CaLH species is formed throughout the pH range, and reaches its maximum metal fraction, corresponding to 0.4, in the range 4 ≤ pH ≤ 10. The stability of complex species in terms of log*K,* referring to reaction (4), is 2.29, indicating rather weak interactions. Increasing the temperature from 25 °C to 37 °C, a decrease in the formation constant value can be observed from 2.29 to 1.68, and a consequent significant reduction of the fraction of the species is evidenced, with a maximum of just over 0.1 in the same pH range.

As widely reported in the literature, the high ability of L to absorb in the UV range has often been used to determine quantitatively and qualitatively this antibiotic in waters [18,42,43]. Various spectrophotometric titrations on solutions containing L and M-L mixtures were performed to investigate L spectral behavior and to confirm the protonation and formation constants values as well as the speciation model obtained by potentiometry, as already shown with other systems [44,45]. The analysis of the UV data, by using the HypSpec program, showed a very good agreement with potentiometric results, highlighted by the similarity of both the values of the protonation constants, as reported in Table 2. The formation constant value of MLH species was also confirmed by UV results. Figure 3 shows, by way of example, some of the spectra acquired on solutions containing L at selected pH in the spectral range 200 ≤ λ ≤ 400 nm. Spectra referring to M-L solutions in the same spectral and pH range show a very similar behavior, as reported in Appendix A.

Figure 3 shows three maximum absorptions at λ = 203, 227, and 319 nm, and points out that all these signals undergo a hyperchromic effect with the increase of pH. An isosbestic point is visible for the LH_2_/LH equilibria. Through spectrophotometric measurements, it was also possible to study the spectral behavior of the species formed in solution. The molar absorbances of the protonated and unprotonated species of the ligand together with ones of the M-L species were reported in Figure 4. In particular, the totally deprotonated species L reaches its maximum value of ε corresponding to 63,000 L mol^−1^ cm^−1^ at λ = 204 nm, while the values of ε reached by the two protonated species LH and LH_2_ are lower (9000 L mol^−1^ cm^−1^ at λ = 319 nm and 4600 L mol^−1^ cm^−1^ a λ = 205 nm, respectively). A second maximum at λ = 319 nm was observed for L species with ε of 10,100 L mol^−1^ cm^−1^. The MLH species reaches its maximum ε value, corresponding to 13000 L mol^−1^ cm^−1^ at λ = 318 nm.

The acid-base behavior of the ligand was also investigated by ^1^H NMR. Various spectroscopic titrations were therefore carried out on solutions containing L to determine the values of the chemical shifts, and to refine the protonation constant values. The ^1^H NMR spectra, shown in Figure 5 at various pH values and *t* = 25 °C, show four distinct signals: a singlet associated with CH_3_ protons, two triplets related to the protons of the two CH_2,_ indicated as CH_2a_ and CH_2b_, and a singlet of the CH proton present in the imidazole ring. An upfield of 0.3 ppm of the CH chemical shift can be observed as a consequence of the deprotonation of the imidazole group for the pH increase. A similar behavior can be observed for the CH_3_ in which the chemical shift is 0.2 ppm. A ^1^H NMR characterization, reported in the literature [46], shows a trend and chemical shift values close to the ones here found. Furthermore, M-L solutions were analyzed by ^1^H NMR at *t* = 25 °C and *I =* 0.15 mol L^−1^, using a procedure already employed for other systems [47,48]. The experimental spectra were reported in the Appendix A. The comparison of these spectra with those obtained on ligand solutions shows that the chemical shift values have the same trend as those recorded for the free ligand, confirming a weak metal-ligand interaction.

### 2.1. Dependence of Protonation and Formation Constants on Ionic Strength

The choice to carry out the experiments at various ionic strengths made it possible to study the dependence of the values of the protonation and formation constants on the ionic strength, expressed by a Debye-Hückel type equation, reported below [49,50]:
(5)$$\log\beta =\log\beta^{T}\;-\;0.51\;z*\frac{\;\sqrt{I}}{1\;+\;1.5\;\sqrt{I}}\;+\;CI$$
where β^T^ is the protonation constant at infinite dilution, z* is the parameter that refers to the charges present in the solution corresponding to *∑*z^2^_reagents_ − *∑*z^2^_products_, and *C* refers to the empirical parameter that depends on the stoichiometric coefficients and on the charges. Using Equation (5) and equilibrium constants at various ionic strength values, the formation constants extrapolated to *I =* 0 mol L^−1^, together with the empirical parameter *C*, were obtained. These values are summarized in Table 3. The knowledge of these parameters made it possible to calculate the value of logβ at any desired ionic strength in the range investigated and therefore for any natural or biological fluid.

### 2.2. Dependence of Protonation and Formation Constants on Temperature

The protonation and formation constants values acquired at the three different temperatures of 15, 25, and 37 °C, listed in Table 1, were used to study their dependence on temperature, as already shown for several systems [51,52]. The data obtained by the potentiometric measurements were analyzed using the following van’t Hoff equation:logβ_T_ = logβ_θ_ + Δ*H*^0^(1/θ − 1/*T*) *R* ln 10(6)
where logβ_T_ is the stability constant at a certain temperature (expressed in kelvin), while logβ_θ_ is the stability constant at *t =* 298.15 K, and R is the universal gas constant expressed as 8.314 J K^−1^ mol^−1^. Using the equation just reported, it was possible to calculate the protonation and formation enthalpies change values, together with Gibbs free energy and entropy changes. The obtained thermodynamic parameters are summarized in Table 4 and shown in Figure 6. As expected, the interaction between Ca^2+^ and L is weak, with a weakly exothermic enthalpy change and a negative entropy change. The protonation equilibria were also characterized by exothermic enthalpy changes.

### 2.3. Sequestering Ability

The study on the interaction of a ligand toward metal cations normally present in natural waters is extremely interesting for the development of systems capable of removing these pollutants from the environment. However, it is known that the mere knowledge of the values of the formation constants is not sufficient for such studies, as it is necessary to consider all the equilibria involved in the analyzed system that could interfere with this interaction. The evaluation of the sequestering ability of L against Ca^2+^, widely present in natural waters, therefore, becomes interesting in order to be able to describe the effectiveness of the system studied. To evaluate this capacity for quantitative purposes, as already widely reported in other previous papers [53,54], the empirical parameter pL_0.5_ was calculated. It represents the total ligand concentration necessary for the sequestration of a mole fraction equal to 0.5 of a specific cation present in trace. The calculation is based on a Boltzmann-type sigmoidal equation:(7)$$\chi =\frac{1}{1+10^{\left(\mathrm{pL}-{\mathrm{pL}}_{0.5}\right)}}$$
where χ is the sum of mole fractions of the different complex species and pL is the anti-logarithm of the total ligand concentration. To evaluate the sequestering ability of the ligands under study toward Ca^2+^, pL_0.5_ values were calculated under different pH and temperatures. Figure 7 shows a comparison of the sequestering ability of L toward Ca^2+^ at different values of ionic strength, at *t* = 25 °C and an average pH of sea water of 8.1. According to the calculated pL_0.5_ values, the following order is:pL_0.5_ (*I* = 0.15 mol L^−1^) > pL_0.5_ (*I* = 1 mol L^−1^) > pL_0.5_ (*I* = 0.7 mol L^−1^)

### 2.4. Speciation in Natural Waters

In the environmental field, it was crucial to assess pollutant speciation and transformation in the aquatic system to improve strategies for its removal from natural waters.

In order to assess the relevance of the Ca^2+^ complex in sea water conditions, the average concentrations of *MNZ* (L), Ca^2+^, Mg^2+^, and other divalent cations such as Cu^2+^ and Zn^2+^, were considered. With the use of the thermodynamic data here determined, it is possible to carry out simulations in sea water conditions. More in detail, the protonation constants of L and the stability constants of Ca^2+^-L complex species were calculated by data in Table 3 at *I* = 0.7 mol L^−1^, i.e., the mean ionic strength of sea water. For Cu^2+^-L and Zn^2+^-L complexes, data of formation constants determined by us and not yet published were used. The simulation was obtained by considering the real concentrations of main divalent cations in sea water, Ca^2+^ and Mg^2+^, and other interfering divalent metal cations, such as Cu^2+^ and Zn^2+^, with L in trace amount (C_L_ = 8·10^−10^ mol L^−1^, C_Ca_ = 0.01 mol L^−1^, C_Mg_ = 0.043 mol L^−1^, C_Cu_ = 4·10^−5^ mol L^−1^, C_Zn_= 1.5·10^−9^ mol L^−1^, *I* = 0.7 mol L^−1^, *t* = 25 °C) [39,55]. Formation constant values of Cu^2+^- and Zn^2+^-L species under sea water conditions, belonging to our unpublished study, are summarized in the Appendix A. Under these conditions, at pH = 8.1 typical of sea waters, a significant percentage equal to 29% of CaLH species was obtained, together with 10% of CuL, 4% of ZnL, 35% of ZnLOH, and 22% of LH (see Figure 8). These results underline that although CaLH species is weak, under sea water conditions, its formation percentage is significant even in the presence of other interfering divalent cations. Therefore, its presence cannot be ignored in the development of L removal strategies.

### 2.5. Comparison with Literature Data

In the literature and in the main thermodynamic databases, the data on *MNZ* (L) protonation constants are few and to our knowledge, no data are present on the Ca-L complexes [56,57,58]. Among them, a protonation constant value of log*K*_2_ = 2.15 of L at *t* = 25 °C and *I* = 0.1 mol L^−1^ in NaClO_4_, obtained by spectrophotometry, was reported in the paper [18]. This value is fairly different from the one here reported with the same technique, log*K*_2_ = 2.47 at *I* = 0.15 mol L^−1^ in NaCl (see Table 2). The difference is attributable to the slight difference in ionic strength and to the different ionic medium. IUPAC stability constants database reports a log*K*_2_ = 2.68 at *t* = 25 °C and *I* = 0.5 mol L^−1^ in KCl obtained by potentiometry [58], very close to the value here reported at the same temperature and ionic strength in NaCl (log*K*_2_ = 2.67, see Table 1). The acid-base study was conducted through potentiometric titrations at 25 °C, *I* = 0.2 mol L^−1^ in NaCl in water–ethanol (90:10, *v*/*v*) mixture, reports three protonation constants values, log*K*_1_ =18.1, log*K*_2_ = 7.01 and log*K*_3_ = 2.64, respectively [46]. The log*K*_3_ value is particularly similar to the one here proposed under the same conditions of temperature and ionic strength, despite the presence of a minimal percentage of methanol.

## 3. Materials and Methods

### 3.1. Materials

Metronidazole and calcium chloride solutions were prepared by weighing and dissolving the commercial product *MNZ* (purity ≥ 97%, Alfa-Aesar/Thermo Fisher, Kandel, Germany) and calcium(II) chloride dihydrate (purity > 99%, Fluka/Honeywell, Charlotte, NC, USA). The exact concentration of the calcium solution was determined by titration with EDTA. The NaOH solutions were prepared, according to the standard procedure, by dilution from the Fluka^®^ concentrated vial and standardized with potassium acid phthalate, previously dried in an oven for at least one hour at a temperature of 110 °C. Similarly, the HCl solutions were prepared by dilution from the Fluka^®^ concentrated solution and standardized with sodium carbonate pre-dried in an oven at 110 °C. The NaCl solutions were often prepared by direct weighing of Sigma-Aldrich^®^ NaCl (purity ≥ 99.5%), first dried in an oven at 110 °C. Distilled water (conductivity < 0.1 μS cm^−1^) and class A glassware were used for all solutions.

### 3.2. Potentiometric Apparatus and Procedure

The potentiometric titrations were employed by Metrohm-Titrando 809 automated potentiometer with a combined glass electrode ORION, type Ross 8102SC and a Metrohm Dosino 800 automatic dispenser. The titrations were performed in cells with a capacity of 25 mL, characterized by a double wall inside which thermostated water circulates. The cells were connected to a thermostat model D1-G Haake. During the measurements, pure N_2_ was bubbled to avoid CO_2_ and O_2_ in solution. The system is interfaced to a PC through the Metrohm TIAMO 2.0 software, which automatically acquires the mL/mV pairs. It can control several parameters, such as e.m.f. stability, titrant delivery, and data acquisition. Estimated accuracy of this apparatus is ±0.15 mV and ±0.002 mL for e.m.f. and for readings of titrant volume. The acid-base titrations were carried out first on 25 mL solutions containing L, HCl, and NaCl as supporting electrolytes at different temperatures and ionic strength conditions. Various potentiometric titrations are performed on 25 mL containing Ca^2+^-L in different ratios, in NaCl at *t* = 15, 25, and 37 °C and *I* = 0.15, 0.5 and 1 mmol L^−1^. The experimental conditions of potentiometric titrations were reported in Table 5. The solutions were titrated using standard NaOH solution. A calibration was associated with each measurement by titration with NaOH of a solution containing HCl, NaCl, and H_2_O, in order to calculate the standard electrode potential (E^0^) and the pKw values.

### 3.3. NMR Apparatus and Procedure

NMR measurements were performed by Varian 500 NMR spectrometer. The ^1^H NMR spectra were recorded using 90% H_2_O and 10% D_2_O as solvent and the chemical shifts were referred to the 1,4-dioxane signal (δ_CHdioxane_ = 3.7 ppm). Titrations were carried out using standard NaOH on 25 mL of solution containing L, NaCl, HCl, 1,4-dioxane and D_2_O. Spectroscopic analyses were performed at selected pH values using 0.6 mL of L solution. Ca^2+^-L spectra were recorded at the same conditions of ionic strength, temperature, and pH range of the free ligand. The experimental conditions of ^1^H NMR titrations were reported in Table 5.

### 3.4. UV-Vis Apparatus and Procedure

The spectrophotometric titrations were performed with a Varian Cary 50 UV-Vis spectrophotometer characterized by an optical fiber with a fixed 1 cm path length. The system was interfaced to a PC by the Varian Cary WinUV software. For all titrations, a Thermo-Fisher glass electrode was also used to record simultaneously the couple of data absorbance (A) and e.m.f. (mV) vs. volume of titrant (mL), for each titration point. The analyses were carried out in a thermostated cell, in which N_2_ was bubbled in solution to avoid CO_2_ and O_2_. The experimental conditions of the titrations are reported in Table 5.

### 3.5. Calculations

The protonation constant of *MNZ* (L), formation constant values of Ca^2+^(M)*-MNZ* (L), and titration parameters (analytical concentration of the reagents, standard potential *E^0^*, junction potential) were determined by using *BSTAC* and *STACO* programs. *LIANA* program has provided the parameters characterizing the dependence of the protonation and formation constant values on the temperature and ionic strength. More details on software are present in ref. [59]. ^1^H NMR data were processed by *HypNMR* software, which allows for calculating protonation and formation constant values and the individual chemical shift of species through the observed signals and assuming fast mutual exchange in the NMR time scale [60]. UV-Vis data are processed by the *HypSpec* program, which allows refining of the protonation and formation constant as well as the molar absorption coefficient values (ε) of the species. The speciation diagrams and the formation percentage of the species are determined by the *Hyss* program [61].

## 4. Conclusions

The main objective of this research was to obtain reliable thermodynamic data capable of describing the environmental behavior of *MNZ* (L)*,* an emerging contaminant present in natural waters, and its interaction with Ca^2+^. Through potentiometric titrations, it was possible to first determine the values of the protonation constants of L, useful for studying the acid-base properties of the ligand, and then exploited to study the complex species formed with Ca^2+^. As expected, the values of the formation constants obtained showed a weak metal-ligand interaction, with a speciation model, characterized by only MLH species, confirmed by UV-Vis and ^1^H NMR analyses. To obtain a complete thermodynamic picture, the dependence on the protonation and formation constants values on the ionic strength and temperature was also determined. The sequestering ability of L towards Ca^2+^ was evaluated as a function of the parameters pL_0.5_ at different pH, ionic strength, and temperature values, with particular attention to those that simulate sea water conditions. The assessment of the thermodynamic data of L and Ca^2+^-L species is fundamental for carrying out simulations and predicting the environmental behavior of these species in real conditions and therefore useful for setting up effective removal processes.

## Figures and Tables

**Figure 1 molecules-27-05394-f001:**
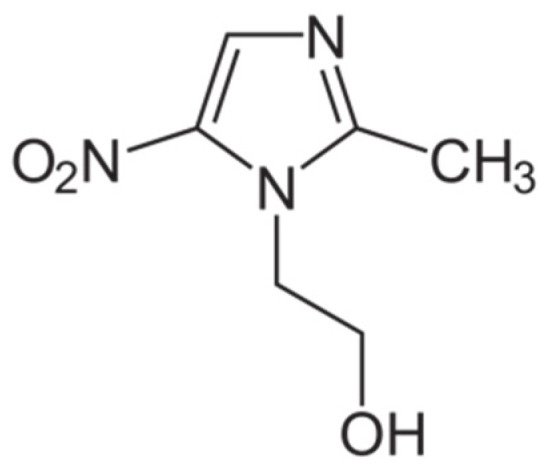
Metronidazole (2-Methyl-5-nitroimidazole-1-ethanol, *MNZ* (L)).

**Figure 2 molecules-27-05394-f002:**
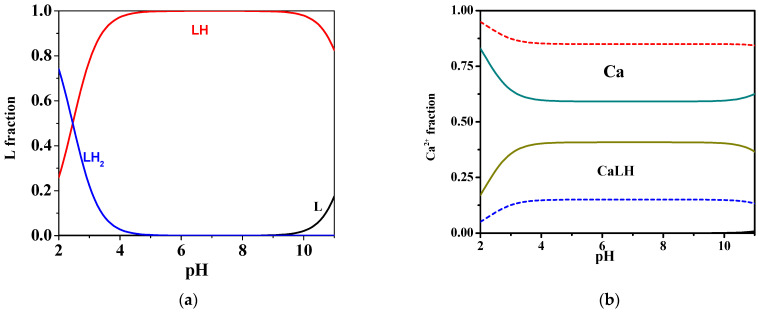
Distribution diagrams at *I =* 0.15 mol L^−1^ in NaCl of: (**a**) *MNZ* (L) system at C_L_ = 2 mmol L^−1^ and *t* = 25 °C; (**b**) Ca^2+^*-MNZ* (L) system at C_L_ = 2 mmol L^−1^, C_Ca_ = 1 mmol L^−1^, *t* = 25 °C (solid lines) and *t =* 37 °C (dotted lines).

**Figure 3 molecules-27-05394-f003:**
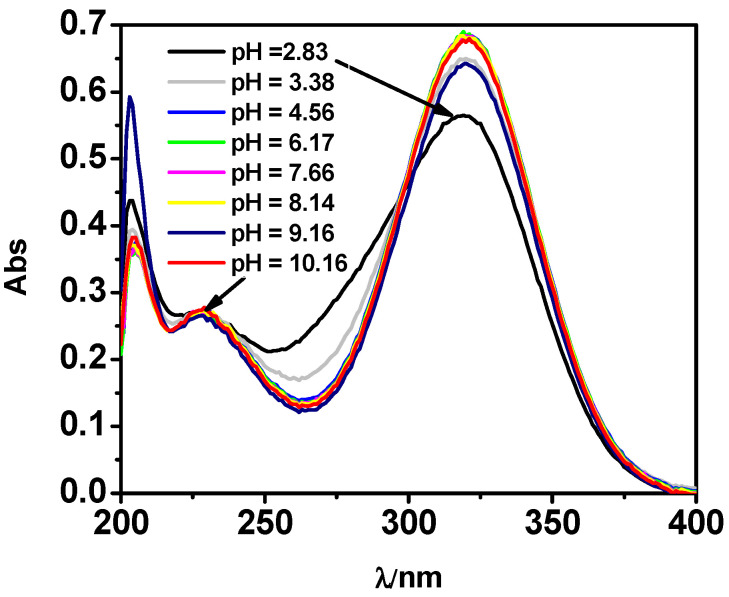
UV spectra at *t* = 25 °C and *I* = 0.15 mol L^−1^ of *MNZ* (L) solution at C_L_ = 0.075 mmol L^−1^.

**Figure 4 molecules-27-05394-f004:**
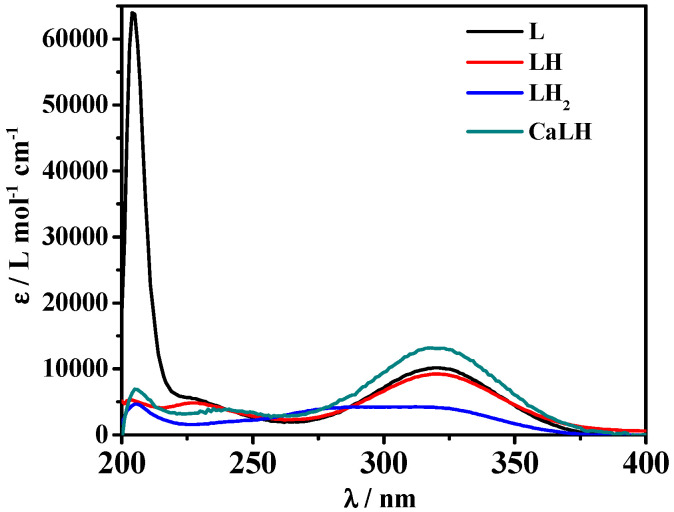
ε vs. λ of Ca^2+^-*MNZ* (L) and *MNZ* (L) species at *t* = 25 °C, *I* = 0.15 mol L^−1^.

**Figure 5 molecules-27-05394-f005:**
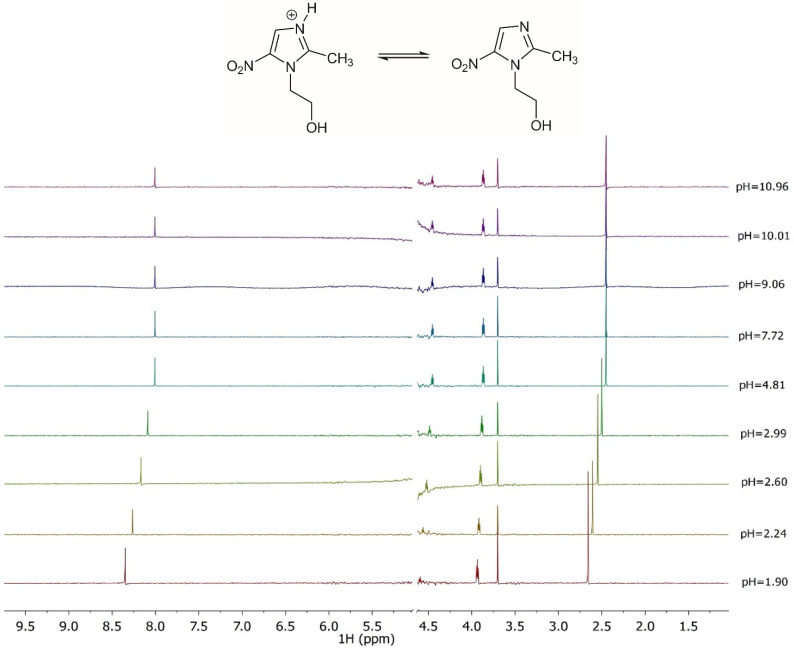
^1^H NMR spectra on solutions containing *MNZ* (L) at C_L_ = 5 mmol L^−1^, *t* = 25 °C, *I* = 0.15 mol L^−1^ in NaCl, 1.90 ≤ pH ≤ 10.96.

**Figure 6 molecules-27-05394-f006:**
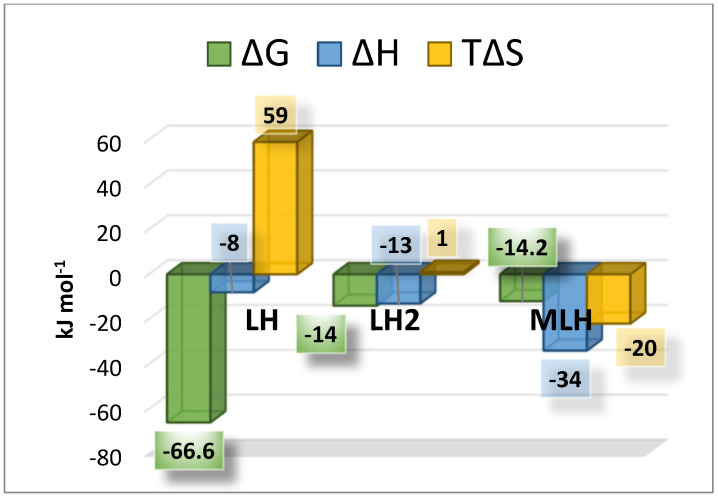
Bar plot of the thermodynamic parameters referred to LH, LH_2_ and MLH species at *t* = 25 °C and *I* = 0.15 mol L^−1^ in NaCl.

**Figure 7 molecules-27-05394-f007:**
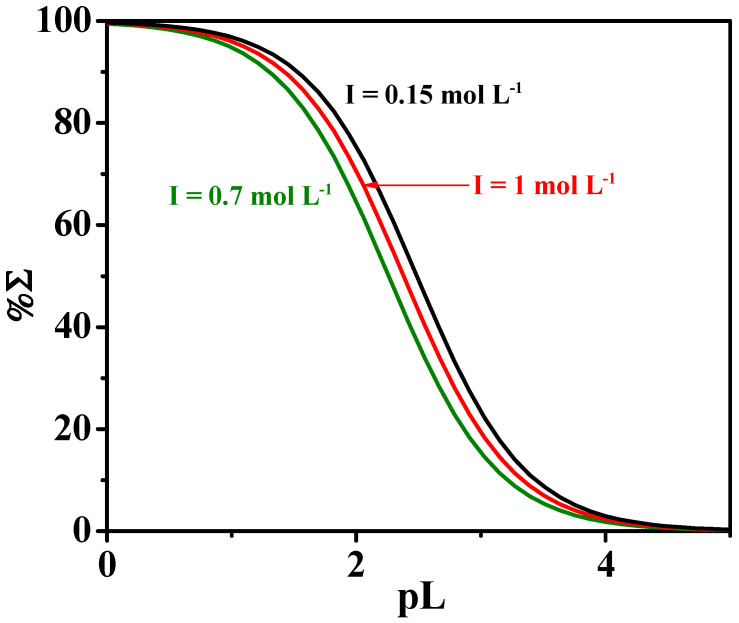
Sequestering ability of *MNZ* (L) toward Ca^2+^ at different ionic strength values, at *t* = 25 °C and pH = 8.1.

**Figure 8 molecules-27-05394-f008:**
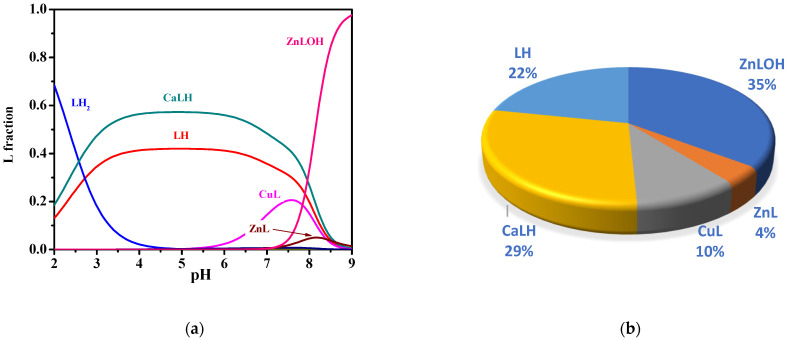
Diagrams of *MNZ* (L) species under sea water conditions (C_L_ = 8·10^−10^ mol L^−1^, C_Ca_ = 0.01 mol L^−1^, C_Mg_ = 0.043 mol L^−1^, C_Cu_ = 4·10^−5^ mol L^−1^, C_Zn_ = 1.5·10^−9^ mol L^−1^, *I* = 0.7 mol L^−1^, *t* = 25 °C): (**a**) Speciation diagram; (**b**) Pie chart at pH = 8.1.

**Table 1 molecules-27-05394-t001:** Experimental protonation constants of *MNZ* (L) and formation constants of Ca^2+^(M)-*MNZ* (L) species obtained by potentiometry.

*t*/°C	*I*/mol L^−1^	logβ^HL 1^	logβ^H2L 2^	logβ^CaLH 3^
15	0.15	12.00(1) ^4^	14.559(5) ^4^	14.06(11) ^4, 5^
	0.5	12.180(8)	14.833(4)	
	0.97	12.12(1)	14.877(7)	
25	0.15	11.674(7)	14.130(4)	14.16(7) ^5^
	0.5	12.51(4)	15.18(4)	15.00(2)
	0.96	12.22(6)	15.05(6)	14.60(5) ^6^
37	0.14	11.88(7)	14.27(8)	13.56(8) ^7^
	0.5	12.14(8)	14.75(8)	
	0.96	12.000(6)	14.601(8)	
***t*/°C**	***I*/mol L^−1^**		**log*K*^H2L 8^**	**log*K*^CaLH 9^**
15	0.15		2.56	2.06 ^5^
	0.5		2.653	
	0.97		2.76	
25	0.15		2.46	2.49 ^5^
	0.5		2.67	2.49
	0.96		2.83	2.38 ^6^
37	0.14		2.39	1.68 ^7^
	0.5		2.61	
	0.96		2.601	

^1^ Refer to the reaction H + L = HL; ^2^ refer to the reaction 2H + L = H_2_L; ^3^ refer to the reaction Ca + L + H = CaLH; ^4^ ≥95% of confidence interval; ^5^ *I* = 0.16 mol L^−1^; ^6^ *I* = 0.98 mol L^−1^; ^7^ *I* = 0.15 mol L^−1^; ^8^ refer to the reaction LH + H = H_2_L; ^9^ refer to the reaction Ca + LH = CaLH.

**Table 2 molecules-27-05394-t002:** Comparison between experimental protonation constants of *MNZ* (L) and formation constant of Ca^2+^*-MNZ* (L) species obtained by potentiometry, ^1^H-NMR and UV spectrophotometry at *t* = 25 °C and *I* = 0.15 mol L^−1^.

Reaction	logβ_H-NMR_	logβ_UV_	logβ_potentiometry_	logβ_average value_
L + H = LH	11.81(1) ^1^	11.73(2) ^1^	11.674(7) ^1^	11.74(2) ^1^
L + 2H = LH_2_	14.13(1)	14.20(6)	14.130(4)	14.15(6)
Ca + L + H = CaLH	-	14.17(2)	14.16(7)	14.16(7)

^1^ ≥95% of confidence interval.

**Table 3 molecules-27-05394-t003:** Protonation constants at infinite dilution and parameters for the dependence on ionic strength (Equation (5)), of *MNZ* (L) and Ca^2+^-*MNZ* (L) species at *t* = 25 °C in NaCl.

Reaction	log^T^β	*C*
L + H = LH	12.0(3) ^1^	0.7(2) ^1^
L + 2H = LH_2_	14.4(1)	1.16(6)
Ca + L + H = CaLH	15.0(2)	1.0(2)

^1^ ≥95% of confidence interval.

**Table 4 molecules-27-05394-t004:** Thermodynamic parameters of *MNZ* (L) and Ca^2+^*-MNZ* (L) species at *t =* 298.15 K and *I =* 0.15 mol L^−1^ in NaCl.

Reaction	Δ*G* ^1^	Δ*H* ^1^	*T*Δ*S*^1^
L + H = LH	−66.6	−8(7)	59
LH + H = LH_2_	−14.0	−13(3)	1
Ca + LH = CaLH	−14.2	−34(8)	−20

^1^ ≥95% of confidence interval.

**Table 5 molecules-27-05394-t005:** Experimental conditions for potentiometric, ^1^H NMR and UV-Vis titrations.

Technique	*t*/°C	I/mol L^−1^	C_M_/mmol L^−1^	C_L_/mmol L^−1^	M/L	pH Range
Potentiometry	15–37	0.15–1	−	1–4	−	2–11
	15–37	0.15–1	1–2	1–4	0.33–2	2–11
^1^H NMR	25	0.15	−	5	−	2–11
	25	0.15	6	5	0.5–0.6	2–11
UV-Vis	25	0.15	−	0.05–0.08	−	2–11
	25	0.15	0.04–0.1	0.05–0.08	0.75–1.5	2–11

## Data Availability

The data presented in this study are available on request from the corresponding author.

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
