# Peer review of "Study on Metronidazole Acid-Base Behavior and Speciation with Ca2+ for Potential Applications in Natural Waters"

_molecules, 2022, doi:10.3390/molecules27175394_

Round 1
Reviewer 1 Report
In the presented manuscript Authors show a study on metronidazole acid-base properties and interaction with Ca2+ ions. The applied methods allow for analyzing the investigated system, however, there are a few significant points that must be re-analyzed or inserted in the paper.
1. Acid-base properties of the ligand
The Authors postulate that the ligand is characterized by two protonation constants: logbeta1=logK1=around 12 and logK2=around 2.5. (The first step involves the protonation of basic nitrogen of imidazolic residue, while the second step refers to the alcoholic residue). In my opinion, based on Fig.2a the reaction of the H+L=HL is related to the protonation of the alcoholic group and then H+HL=H2L is related to the (imidazole nitrogen is more acidic than the alcoholic group). However, the values are significantly different from the literature (pKaIm is around 6-7 and pKa of alcoholic group higher than 15) and should be discussed in the paper.
2. The presentation of the Ca2+ complexes in the presence of Cu2+ and Zn2+ ions, without presentation of the results for the Ligand-Cu2+ and ligand Zn2+ with one sentence "For Cu2+- MNZ and Zn2+- 232 MNZ complexes, is not complete. Results for Cu2+ and Zn2+ ions must be inserted in the paper.
Author Response
In the presented manuscript Authors show a study on metronidazole acid-base properties and interaction with Ca2+ ions. The applied methods allow for analyzing the investigated system, however, there are a few significant points that must be re-analyzed or inserted in the paper.
We thank the reviewer for his/her comments and the very careful review.
- Acid-base properties of the ligand
The Authors postulate that the ligand is characterized by two protonation constants: logbeta1=logK1=around 12 and logK2=around 2.5. (The first step involves the protonation of basic nitrogen of imidazolic residue, while the second step refers to the alcoholic residue). In my opinion, based on Fig.2a the reaction of the H+L=HL is related to the protonation of the alcoholic group and then H+HL=H2L is related to the (imidazole nitrogen is more acidic than the alcoholic group). However, the values are significantly different from the literature (pKaIm is around 6-7 and pKa of alcoholic group higher than 15) and should be discussed in the paper.
Unfortunately, there had been an oversight in the manuscript that we hadn't noticed. We thanks very much the reviewer for noticing and pointing this out. Indeed, the first step involves the protonation of alcoholic group, and the second step refers to the imidazolic residue, as indicated in the paper of Rivera-Utrilla et al. Removal of Nitroimidazole Antibiotics from Water by Adsorption/Bioadsortion on Activated Carbon and Advanced Oxidation Processes. Trends in Chemical Engineering 12 (2009): 51-69. We have corrected this mistake in the manuscript.
As concerns the comment on protonation constant values, it must be specified that the protonation constants reported by the reviewer are referred to pure alcohol and imidazole (pKaIm is around 6-7 and pKa of alcoholic group higher than 15). In the case of the ligand under study, 2-Methyl-5-nitroimidazole-1-ethanol, the protonation constants are significantly affected by the presence of other groups. Furthermore, the mean value of pKaim, reported by us (logK2 = 2.41 at t = 25°C, I = 0.15 mol L-1 in NaCl, see Table 2) is confirmed by three different techniques and it is coherent with value reported in the literature (Pettit, L.D., and K.J. Powell. Iupac Stability Constants Database.: Academic Software, IUPAC, 2001; Bouchoucha, A., A. Terbouche, M. Zaouani, F. Derridj, and S. Djebbar. "Iron and Nickel Complexes with Heterocyclic Ligands: Stability, Synthesis, Spectral Characterization, Antimicrobial Activity, Acute and Subacute Toxicity." J. Trace Elements Medicine Biology 27 (2013): 191-202).
- The presentation of the Ca2+ complexes in the presence of Cu2+ and Zn2+ ions, without presentation of the results for the Ligand-Cu2+ and ligand Zn2+ with one sentence "For Cu2+- MNZ and Zn2+- 232 MNZ complexes, is not complete. Results for Cu2+ and Zn2+ ions must be inserted in the paper.
We have included in Supplementary Material a table with the formation constant values of Cu2+-MNZ and Zn2+-MNZ systems under sea water conditions. These values, obtained by us by experimental measurements, have not yet been published.
Reviewer 2 Report
The manuscript by Giuffre et al reports the speciation studies in the calcium2+ – metronidazole (MNZ) – H+ system. Metronidazole is known to be a widely used antibiotic drug. As a result of this, metronidazole is now becoming a new pollutant for nature. In this context, the system the authors took for their speciation studies allowed them to model the behaviour of metronidazole in natural waters. The authors have obtained a large body of self-consistent data on the protonation constants of MNZ and the stability constants of calcium-metronidazole species by potentiometric, 1H NMR and UV-vis titrations. They were able to put their wealth of data in a proper context by comparing them with literature data in the section 3.5. I appreciate that the authors have paid special attention to the estimation of uncertainties of the reported thermodynamic values in their manuscript. The manuscript is well written and organized. The data reported in it will be of interest for analytical chemists and specialists in environmental chemistry. Overall, I find this highly professionally written manuscript suitable for the Analytical Chemistry section of Molecules. I am glad to recommend this manuscript for publication subject to the following revisions.
1) Could you check the Debye-Huckel type equation 5? Shouldn’t the expression (1+1.5 I0.5) be in the denominator?
2) I find it a bit difficult to switch between similar abbreviations, MNZ, MLH, CaMNZH, etc. when reading the manuscript difficult to switch between them when reading the manuscript. What is more confusing is that the authors use the abbreviations MNZ and L interchangeably for metronidazole. I would recommend the authors to elaborate a new easy to follow system of abbreviations and use it throughout the text. For example, when the authors discuss the protonation or the complexation of metronidazole, it would be better to use the abbreviation L in all cases, not MNZ.
3) Lines 253-257. In view of the difference between the logK2 data obtained by the authors and the data reported in ref. [18] I would encourage the authors to check whether this difference can be attributed to the differences in the ionic strength and different ionic medium (NaClO4 vs NaCl).
4) Line 298. The title should read “4.3. 1H NMR Apparatus and Procedure”.
5) Table 1. I have not found the meaning of the footnote “a)” in Table 1 for the logB values for the following equilibria: L+2H=LH2, t=15 C, I = 0.15 and Ca+L+H=CaLH, t=15 C, I = 0.16.
6) Figure 3. It is difficult for me to follow the changes in the absorption at different pH in Figure 3. Could you modify colours in this figure and mark the initial and final spectra recorded at pH = 2.83 and 10.16 with arrows?
7) Lines 92 – 95. I would recommend the authors to briefly discuss the alternative models of speciation in Supplementary Information.
8) Table 4. Why are the dG values are given without their errors? Shouldn’t they have only two significant figures in view of the errors in the dH and TdS terms? The units should also be given for the data in this table.
Author Response
The manuscript by Giuffre et al reports the speciation studies in the calcium2+ – metronidazole (MNZ) – H+ system. Metronidazole is known to be a widely used antibiotic drug. As a result of this, metronidazole is now becoming a new pollutant for nature. In this context, the system the authors took for their speciation studies allowed them to model the behaviour of metronidazole in natural waters. The authors have obtained a large body of self-consistent data on the protonation constants of MNZ and the stability constants of calcium-metronidazole species by potentiometric, 1H NMR and UV-vis titrations. They were able to put their wealth of data in a proper context by comparing them with literature data in the section 3.5. I appreciate that the authors have paid special attention to the estimation of uncertainties of the reported thermodynamic values in their manuscript. The manuscript is well written and organized. The data reported in it will be of interest for analytical chemists and specialists in environmental chemistry. Overall, I find this highly professionally written manuscript suitable for the Analytical Chemistry section of Molecules. I am glad to recommend this manuscript for publication subject to the following revisions.
We thank the reviewer for his/her positive judgment and careful review. We have tried to follow his/her indications as much as possible to correct the manuscript.
1) Could you check the Debye-Huckel type equation 5? Shouldn’t the expression (1+1.5 I0.5) be in the denominator?
The expression (1+1.5 I0.5) is preceded by the symbol "/". However, it was not clearly visible in the equation. Therefore, the equation has been rewritten more clearly indicating its presence in the denominator.
2) I find it a bit difficult to switch between similar abbreviations, MNZ, MLH, CaMNZH, etc. when reading the manuscript difficult to switch between them when reading the manuscript. What is more confusing is that the authors use the abbreviations MNZ and L interchangeably for metronidazole. I would recommend the authors to elaborate a new easy to follow system of abbreviations and use it throughout the text. For example, when the authors discuss the protonation or the complexation of metronidazole, it would be better to use the abbreviation L in all cases, not MNZ.
We included the abbreviation L for metronidazole throughout the manuscript as suggested by the reviewer.
3) Lines 253-257. In view of the difference between the logK2 data obtained by the authors and the data reported in ref. [18] I would encourage the authors to check whether this difference can be attributed to the differences in the ionic strength and different ionic medium (NaClO4 vs NaCl).
The difference between our value and that of Abdel-Kadel et al. is quite significant and it is certainly attribute to the different ionic strength and, mostly, ionic media. We have already commented on this in the manuscript but further comments would require a more in-depth study on the effect of the ionic medium on protonation equilibria.
4) Line 298. The title should read “4.3. 1H NMR Apparatus and Procedure”.
Ok. The title is already like that.
5) Table 1. I have not found the meaning of the footnote “a)” in Table 1 for the logB values for the following equilibria: L+2H=LH2, t=15 C, I = 0.15 and Ca+L+H=CaLH, t=15 C, I = 0.16.
We apologize for the mistake and have corrected footnote and resized Table 1.
6) Figure 3. It is difficult for me to follow the changes in the absorption at different pH in Figure 3. Could you modify colours in this figure and mark the initial and final spectra recorded at pH = 2.83 and 10.16 with arrows?
We are sorry for the difficulties encountered by the reviewer in interpreting Figure 3. We have changed the colors of the lines and indicated with arrows the lines corresponding to the extreme pH values.
7) Lines 92 – 95. I would recommend the authors to briefly discuss the alternative models of speciation in Supplementary Information.
We are sorry for not making this point raised by the reviewer clearly in the manuscript, but we have no alternative models of speciation. Although we have tried to calculate the constants of other species, the only model that produced results was the one already reported in the manuscript and i.e. with the single CaLH species.
8) Table 4. Why are the dG values are given without their errors? Shouldn’t they have only two significant figures in view of the errors in the dH and TdS terms? The units should also be given for the data in this table.
We did not insert the error in the DG values as these are values calculated from the logK value by the equation DG = -RTlnK. The significant figures are more than two since DG values were calculated from the logK values, affected by very low errors. We removed the error from the TDS values as they are also values calculated by the equation TDS = DH - DG.
Table 1 could be rearranged to be shorter.
Table 1 has been reduced.
Reviewer 3 Report
This work is well conducted and written, but I think this article does not present enough scientific novelty and repercussion to be published in the journal Molecules.
As improvements to be introduced in the article, almost all formal, I would highlight the need for a thorough revision of double spaces, the homogenization of units (ml or mL, as an example) and the improvement of the quality of the figures, such as Fig. 6 and 8B.
There are some errors in the introduction regarding the analytical methods. HPLC does not analyse, it separates, so it has to be coupled to a detector if used for analytical purposes, as the authors have indicated in HPLC-MS/MS. In the cited article on biosensors, there is no determination of the MNZ, so it should be deleted.
Table 1 could be rearranged to be shorter.
Author Response
This work is well conducted and written, but I think this article does not present enough scientific novelty and repercussion to be published in the journal Molecules.
We thank the reviewer for the initial positive judgment. We are sorry that in his/her opinion the manuscript does not present enough novelty. As it was described in the manuscript, metronidazole is now becoming a new pollutant in aquatic systems and our study aims to model its behaviour in this environment. We think that our data can be of considerable help to understand the environmental fate of this compound and to develop suitable removal procedures.
We tried to follow all his/her advice to correct the manuscript.
As improvements to be introduced in the article, almost all formal, I would highlight the need for a thorough revision of double spaces, the homogenization of units (ml or mL, as an example) and the improvement of the quality of the figures, such as Fig. 6 and 8B.
We tried to homogenize the manuscript as much as possible and to improve the quality of the figures.
There are some errors in the introduction regarding the analytical methods. HPLC does not analyse, it separates, so it has to be coupled to a detector if used for analytical purposes, as the authors have indicated in HPLC-MS/MS. In the cited article on biosensors, there is no determination of the MNZ, so it should be deleted.
We thank the reviewer for reporting the error inherent the manuscript on biosensors. The citation has been deleted.
Table 1 could be rearranged to be shorter.
Table 1 has been reduced.